# BRACNAC: A *BRCA1* and *BRCA2* Copy Number Alteration Caller from Next-Generation Sequencing Data

**DOI:** 10.3390/ijms242316630

**Published:** 2023-11-22

**Authors:** Andrey Kechin, Ulyana Boyarskikh, Viktoriya Borobova, Evgeniy Khrapov, Sergey Subbotin, Maxim Filipenko

**Affiliations:** 1Institute of Chemical Biology and Fundamental Medicine, Novosibirsk 630090, Russia; 2Faculty of Natural Sciences, Novosibirsk State University, Novosibirsk 630090, Russia

**Keywords:** *BRCA1*, *BRCA2*, CNV, large rearrangements, copy number variations, NGS, targeted sequencing, bioinformatics tool

## Abstract

Detecting copy number variations (CNVs) and alterations (CNAs) in the *BRCA1* and *BRCA2* genes is essential for testing patients for targeted therapy applicability. However, the available bioinformatics tools were initially designed for identifying CNVs/CNAs in whole-genome or -exome (WES) NGS data or targeted NGS data without adaptation to the *BRCA1/2* genes. Most of these tools were tested on sample cohorts of limited size, with their use restricted to specific library preparation kits or sequencing platforms. We developed BRACNAC, a new tool for detecting CNVs and CNAs in the *BRCA1* and *BRCA2* genes in NGS data of different origin. The underlying mechanism of this tool involves various coverage normalization steps complemented by CNV probability evaluation. We estimated the sensitivity and specificity of our tool to be 100% and 94%, respectively, with an area under the curve (AUC) of 94%. The estimation was performed using the NGS data obtained from 213 ovarian and prostate cancer samples tested with in-house and commercially available library preparation kits and additionally using multiplex ligation-dependent probe amplification (MLPA) (12 CNV-positive samples). Using freely available WES and targeted NGS data from other research groups, we demonstrated that BRACNAC could also be used for these two types of data, with an AUC of up to 99.9%. In addition, we determined the limitations of the tool in terms of the minimum number of samples per NGS run (≥20 samples) and the minimum expected percentage of CNV-negative samples (≥80%). We expect that our findings will improve the efficacy of *BRCA1/2* diagnostics. BRACNAC is freely available at the GitHub server.

## 1. Introduction

The clinical significance of *BRCA1* and *BRCA2* pathogenic germline variants and somatic alterations is of great importance for patients with breast and ovarian cancer. Their detection allows at-risk patients to be identified and more effective treatment strategies to be chosen, such as PARP inhibitors for carriers of pathogenic variants. Numerous commercial tests were developed to detect point and short mutations. These include the AmpliSeq BRCA Panel (Illumina, San Diego, CA, USA), the GeneRead QIAact BRCA 1/2 Panel (Qiagen, Hilden, Germany), the Ion AmpliSeq™ *BRCA1* and *BRCA2* Panel (Thermo Fisher Scientific, Waltham, MA, USA), and the ACCEL-AMPLICON^®^
*BRCA1* and *BRCA2* Panel (Swift Biosciences, Ann Arbor, MI, USA). These developments improved the statistics of the occurrence of pathogenic *BRCA1/2* variants worldwide and enlarged the patient groups suitable for targeted therapy. Moreover, somatic large rearrangements (LRs) are considered to be involved in the mechanism of drug resistance [1], which further increases their clinical relevance.

However, the study of large rearrangements, including germline copy number variations (CNVs) and somatic copy number alterations (CNAs), remains limited due to the scarcity of reliable programs that can be tested on a large sample of patients [2,3]. In some studies, large rearrangements (LRs) were detected by combining several methods, including NGS dosage, microarray comparative genomic hybridization (CGH), and/or multiplex ligation-dependent probe amplification (MLPA) [4]. The latter has long been considered the gold standard for CNV detection. The methodology involves the amplification of probes that can be ligated only if they both hybridize to neighboring regions, yielding PCR products with different lengths for multiple target regions [5]. It would be more significant to detect CNVs and CNAs along with short mutations routinely identified with targeted NGS. Several new algorithms for CNV detection were developed for targeted NGS data, with some organized into bioinformatic tools, e.g., panelcn.MOPS [6], CNVpytor [7], and several simple R scripts [8]. All of them use coverage depth values for target regions, detecting outlier values, i.e., the signals of potential CNVs. At the same time, these algorithms have some differences. For example, the R scripts developed by Singh et al. [8] create a pool of normal samples, evaluate the coverage for many sliding windows, and calculate the ratio of coverage depth for the query sample and a pool of normal samples for each nucleotide of the sliding windows. Panelcn.MOPS additionally applies one-step normalization to the third quartile, also selecting control samples. The unique feature of CNVpytor is the correction of the GC content and the use of variant allele frequency (VAF) values. Despite the high sensitivity and specificity exhibited by the tools mentioned above for the datasets tested, their suitability for other datasets and in-house targeted NGS panels remains unverified in large cohorts, particularly for the two genes *BRCA1* and *BRCA2*, with frequent CNVs.

Here, we present a new tool for the detection of *BRCA1/2* CNVs, called BRACNAC (BRAC1/2 Copy Number Alteration Caller). It should be noted that our tool does not require a control sample set. The method involves several steps of data normalization and the identification of sets of target regions with elevated or lowered copy numbers. Of critical relevance is the estimation of the probability of an observed copy number increase or decrease to be random, i.e., the *p*-value. We compared the performance of our tool with those of the multiplex ligation-dependent probe amplification (MLPA) method regarded as the gold standard for CNV detection and of another recently reported tool for targeted NGS panels, referred to as panelcn.MOPS. We confirmed the suitability of our tool for targeted NGS data from other studies and whole-exome (WES) sequencing data. Furthermore, we show that many *BRCA1/2* NGS datasets previously described could be reanalyzed using our tool to extend our knowledge of *BRCA1/2* CNV occurrence. Finally, we describe our attempt to define the limits of BRACNAC by identifying CNVs with different numbers of samples per run and different ratios of patients with and without CNVs.

## 2. Results

The BRACNAC algorithm and initial values of different parameters were chosen based on the in-house dataset of 213 leukocyte DNA samples. The whole coding sequences of the *BRCA1* and *BRCA2* genes were obtained with in-house and commercial targeted NGS panels. Other datasets were used for comparison with the panelcn.MOPS results and validation on other NGS library enrichment approaches and sequencing technologies (Table 1).

### 2.1. Algorithm

The main idea of the BRACNAC algorithm is to remove any potential variation between samples (first normalization step), between target regions (second normalization step), and arising from differences in multiplex reactions (third (optional) normalization) or clusters of sample sets (e.g., samples whose libraries were prepared in different experiments) (Figure 1A–D). The normalization steps are carried out by dividing each value by the median value of the corresponding sets of values. The resulting value should be 1.0 if there are no changes in copy number for a target. Therefore, by multiplying the resulting value by 2, we should obtain the number of copies of the target in the diploid genome. Then, the algorithm searches for values that deviate from 2.0. The theoretical values for deletion and duplication are 1.0 and 3.0, respectively. In the case of CNVs/CNAs of several exons or one exon but covered by several target regions, these deviating values follow each other, forming a continuous set. However, natural variation in the number of reads makes these values vary around 1.0 and 3.0, respectively. Exons comprising several target regions may demonstrate values with lower deviation than that obtained for one copy (e.g., 1.3 instead of 1.0 for deletions and 2.7 instead of 3.0 for amplifications). Therefore, the initial algorithm allows determining four thresholds to consider some coverage value of a target region for deletions or amplifications. The second values (higher for deletions, “del2”, e.g., 1.7 and lower for amplifications, “dupl2”, e.g., 2.4) are used for calculating the CNV/CNA score. The first values (lower for deletions, “del1”, e.g., 1.3 and higher for amplifications, “dupl1”, e.g., 2.7) are used for lowering the score if some values are between the first and the second thresholds. We set the limits that any potential CNV/CNA score should include at least one value less than “del1” (for deletions) or higher than “dupl1” (for amplifications) and no more than one value higher than “del2” (for deletions) or lower than “dupl2” (for amplifications) (Figure 1E,F). Additionally, the lower threshold of deletions is used as the basis of the power function for decreasing the score (1.3 in the equations of the Section 4).

We collected different overlapping sets of target regions by extracting the normalized coverage values of neighboring target regions exhibiting copy numbers below 2.0 (for deletions) or above 2.0 (for amplifications). Subsequently, we calculated the scores and estimated the *p*-values based on the probability of a given set of target regions to have such a score at random. For that purpose, the normalized coverage values of the analyzed sample were randomly shuffled. Then, this procedure was repeated for the values of the same target regions of other samples. In this way, we accounted for variation of the coverage values of the examined sample and target regions among all samples. The *p*-value evaluation by the procedure described above was performed in two steps. In the first step, BRACNAC identified CNVs/CNAs with a high score (by default, 9.9) and a low *p*-value (by default, 0.01). Then, it excluded such samples from the second step of *p*-value evaluation to avoid their influence on the probability values for other potential CNVs/CNAs. We implemented this two-step *p*-value evaluation procedure to decrease the false-negative rate.

### 2.2. BRACNAC vs. MLPA and Panelcn.MOPS for Targeted NGS

Two hundred eleven leukocyte DNA samples from ovarian cancer patients and two samples from prostate cancer patients were analyzed using targeted NGS, followed by BRACNAC and panelcn.MOPS analysis and MLPA. MLPA identified twelve positive cases (Table 2). The coverage values are provided in Appendix A, and the ratio plots are shown in Appendix A. An example of BRACNAC output plot for a positive sample is shown in Figure 2.

For BRACNAC, several sets of parameters were tested (Figure 3). CNVs with a match between mutation type and gene (*BRCA1* or *BRCA2*) were considered true positive, with the remaining cases deemed to be false positive. Therefore, four (if *BRCA2* was also tested with MLPA) or two comparisons were performed for each patient. Unfortunately, the low number of positive cases prevented us from performing fair tests (subdividing the samples into training and test groups) and ROC analysis for deletions and amplifications, separately. However, even with no optimization of the BRACNAC parameters, its performance yielded an area under the curve (AUC) of more than 94% (Figure 3), with a sensitivity of 92–100% and a specificity of 94%.

Panelcn.MOPS testing of the resulting NGS data identified all the samples with CNVs. However, there were a lot of false-positive exon deletions/duplications for the true-positive samples and 178 false-positive samples, yielding a sensitivity of 100% and a specificity of 11%. The percentage of complete matches for the CNV-positive samples was 58%. Provided that the clinical values were based on the number of correctly determined copies of exons, the sensitivity and specificity were 64 and 84%, respectively. The accuracy of CNV detection did not correlate with the NGS assay used (Qiagen or in-house assay).

The panelcn.MOPS data included only three *BRCA1/2* CNV-positive and 96 CNV-negative samples. BRACNAC and panelcn.MOPS identified all positive samples correctly. BRACNAC detected six false-positive CNVs but only one with *p*-value = 0.006, which was observed for all true-positive samples. Thus, the AUC for BRACNAC and panelcn.MOPS was 99.9% and 100%, respectively.

To confirm that BRACNAC can also be used for other targeted NGS panels and NGS platforms, we applied it to call CNVs from the AmpliSeq *BRCA1* and *BRCA2* NGS Panel and the Ion Proton platform data. The procedure involved 192 leukocyte DNA samples. The BRACNAC excluded eight samples due to a low median coverage (less than 100). We detected seven *BRCA1* exon deletions and two exon amplifications, respectively, with a *p*-value of ≤0.001 (Table 3 and Appendix A). During manual figure analysis, at least two CNVs identified (*BRCA1* ex2-ex22 and *BRCA1* ex23-ex23 deletions) could be considered true-positive (see Appendix A). These two patients were also diagnosed at early ages (33 and 35). Other cases should be confirmed by an alternative assay.

### 2.3. BRACNAC for WES

To confirm that BRACNAC can also be used for WES NGS data, we used it to call CNVs from the available data, although they were poorly described. The database included 60 FFPE DNA samples containing germline *BRCA1/2* mutations (Table 4). Large deletions and amplifications were found for eight and five patients, respectively. Based on the BRACNAC plots (Appendix A), we can suggest that the SRR5604273 and SRR5604295 samples contained germline CNVs, and the others only somatic CNAs.

For the SRR5604279 sample, a CNV was identified as a deletion of *BRCA1* exons 17–19. However, we suggest the complete *BRCA1* deletion for this sample to be due to a possible low tumor cell percent. For the SRR5604275 and SRR5604299 samples, deletion of the whole *BRCA1* was identified.

### 2.4. BRACNAC Limitations

To identify the limitations of BRACNAC, we applied it to the in-house data of patients for which MLPA analysis had been performed. We tested three types of limits: (1) a minimal number of samples per NGS run; (2) a maximally acceptable percentage of *BRCA1/2* CNV-positive samples per random set tested (with at least one CNV-positive sample per dataset); (3) the opportunity to use several NGS run data for the same library preparation kit. First, we tested BRACNAC on 100 random sets of samples with up to 70 patients from one (Figure 4a) or several (Figure 4b) NGS runs. At least six samples were necessary to obtain an AUC higher than 80%. Furthermore, using several runs of NGS data considerably decreased the CNV detection efficacy, especially for the sets with more than 60 samples. To determine which maximal percentage of CNV-positive samples per NGS dataset was acceptable for accurate CNV detection, we tested BRACNAC on 100 random sets of 100 samples with a share of CNV-positive samples from 0.1 to 0.9 (Figure 4c). We combined the NGS data of several NGS runs due to the low number of CNV-positive samples (12) that led to low AUCs. However, we identified a rapid decline in the AUC value for sets with more than 40% of CNV-positive samples.

## 3. Discussion

This work described a new *BRCA1/2* CNV/CNA detection tool and compared it with the freely available program panelcn.MOPS [6] and the gold standard MLPA. We obtained high sensitivity (100%) and specificity (94%) values. In many cases, MLPA confirmation and clarification of rearrangement boundaries was necessary, as in other studies [5]. We believe that even this two-step CNV detection (NGS + MLPA) could significantly simplify the process of *BRCA1/2* LR identification. Compared with panelcn.MOPS, we observed many false-positive CNVs and CNVs with boundaries incorrectly determined by panelcn.MOPS when using the in-house amplicon-based targeted NGS data. These results were likely due to the fact that the normalized copy numbers for each amplified region were not taken into account when evaluating the probability of whole exon deletion or duplication performed with BRACNAC. In contrast, panelcn.MOPS showed a slightly better performance for the hybridization-based targeted NGS data.

We also demonstrated that BRACNAC could be applied to WES data. However, we could not compare our method with any alternative assay, and the coverage depth required can be higher than the usual one. Another limitation of using BRACNAC for WES data is the absence of non-coding exon coverage. A high frequency of *BRCA1* exons 1–2 deletion was shown in our study (25% of all CNVs confirmed with MLPA) and other investigations [10]. Thus, the inclusion of exons 1–2 and a promoter region may improve the detection of such CNVs. Another improvement for in-house and commercial NGS assays could be using additional amplicons for short exons and flanking introns covered by only one amplicon, potentially increasing the accuracy of such exon CNV detection.

One of the limitations of BRACNAC is the inability to detect automatically the partial CNVs of exons with only some part of the exon deleted or amplified. However, the operator could suggest some CNVs based on the BRACNAC ratio plots. Other limitations discovered in this study can be extended to similar CNV detection tools, such as panelcn.MOPS. We found that it is inadvisable to combine the data from multiple runs, as it could lead to a decreased accuracy of the detection of large rearrangements. Furthermore, we demonstrated that the minimally acceptable number of samples per single NGS run is 20 samples with at least 80% of CNV-negative samples. This percentage is significantly lower than the expected CNV occurrence among breast and ovarian cancer patients in different populations, which varies from 2 to 10% depending on the race of the patients and the sample size [11,12]. This result allows one to use the NGS data of sample cohorts that were not selected for CNV presence in the genome.

Although the sample cohort used for evaluating BRACNAC is one of the largest described in the literature (e.g., panelcn.MOPS was evaluated on 180 samples) [6,13], we could not estimate a fair frequency of CNVs in our cohort due to the non-random selection of samples to confirm the CNV status by MLPA. We plan to perform such an evaluation in the future.

Another promising direction for further research is the comparison of existing programs for identifying somatic CNAs. However, the absence of a gold standard complicates such a study. For example, MLPA is highly sensitive to tissue fixation conditions [14] as well as ddPCR, but this sensitivity is less pronounced for the latter due to the absence of the ligation step. Therefore, ddPCR is a potential method of choice that could be applied to blood and FFPE DNA samples [15,16].

## 4. Materials and Methods

### 4.1. Algorithm

BRACNAC is an open-source and freely available Python tool that includes several parts for effectively calling CNVs and CNAs in the *BRCA1* and *BRCA2* genes (Figure 5). It uses the target region coordinate file and TSV table with samples’ coverage depth values for the target regions. The coverage values for each target region can be mean, median, minimal, or maximal coverage values per all nucleotides of the region, but they should be uniform across all the samples analyzed. In the case of amplicon-based targeted NGS panels, such regions can be represented by the genomic coordinates of the amplified regions. From the input table, BRACNAC creates a NumPy array that is normalized by the following types: (1) the median coverage values of one sample for the target regions; (2) the median normalized coverage value for each target region among all samples; (3) the median normalized coverage value for each primer pool (if it is known or applicable, e.g., for an amplicon-based targeted NGS panel); (4) the median normalized coverage for each sample cluster (optional, applicable, e.g., when some sample libraries are prepared separately from the rest). To obtain target region copy numbers, BRACNAC multiplies all values by 2.

After the normalization steps, BRACNAC searches for exons potentially deleted and/or amplified: (1) it combines the values less than 2.0 and greater than 2.0 to two distinct sets; (2) it joins the neighboring value ranges if the region between them has low coverage or has a value less than 2.0 (for deletions) or greater than 2.0 (for amplifications); (3) it filters out the value ranges where all the exons are affected only partially (this filtration is performed because almost all *BRCA1/2* LRs are associated with Alu-repeats in their introns); (4) it filters out the deletions for which the number of normalized values of at least of 1.3 (default value, but can be changed by the “del1” parameter) is less than one; (5) it filters out the deletions for which the number of values higher than 1.7 (“del2” parameter) is greater than one; (6) it filters out the deletions for which the number of values of no more than 1.7 (“del2” parameter) is less than half of the length of the region; (7) it filters out the amplifications for which the number of normalized values of at least 2.7 (default value which can be changed by the “dupl1” parameter) is less than one; (8) it filters out the amplifications for which the number of values less than 2.4 (“dupl2” parameter) is greater than one; (9) it filters out the amplifications for which the number of values of at least 2.4 (“dupl2” parameter) is less than half of the length of the region; (10) for the left deletions and amplifications, it calculates the score and *p*-values and filters them out by these values.
The deletion score is calculated as follows:
(1)$${score}_{del}=\frac{k_{1}\times \sum_{i=1}^{N}\left(4-a_{i}\right)-4\times k_{2}-4\times k_{3}}{1.3^{\left(\frac{D}{L}+k_{4}\times 2+\frac{1}{E}+Mdist\right)}}$$
where *k*_1_ is 1 if all considered exons are affected completely or 0.5 if any exon is involved into CNVs/CNAs only partially; *a_i_* indicates normalized target region coverage values that are no more than “del2”; *N* is the number of such values; *k*_2_ is 1 if the first target region value is no greater than “del2”—otherwise it is 0.5; *k*_3_ is 1 if the last target region value is no greater than ”del2”—otherwise it is 0.5; *D* is the number of target region values which are greater than ”del1”; *L* is the number of target region values that are considered as potential deletions; *k*_4_ has a value of 1 if the exon deletion in question is covered by only one target region (except for the second exon, which is frequently deleted in *BRCA1* with promoter regions)—otherwise, *k*_4_ has a value of 0; *E* is the number of exons affected; *Mdist* is the median for the values of the differences between neighboring normalized coverage values.
The amplification score is calculated as follows:
(2)$${score}_{amp}=\frac{k_{1}\times \sum_{i=1}^{N}a_{i}-4\times k_{2}-4\times k_{3}}{1.3^{\left(\frac{I}{L}+k_{4}\times 2+\frac{1}{E}+Mdist\right)}}$$
where *k*_1_ is 1 if all considered exons are affected completely, or 0.5 if any exon is affected only partially; *a_i_* indicates normalized target region coverage values that are at least “dupl2”; *N* is the number of such values; *k*_2_ is 1 if the first target region value is at least “dupl2”—otherwise, it is 0.5; *k*_3_ is 1 if the last target region value is at least “dupl2”—otherwise, it is 0.5; *I* is the number of target region values less than “dupl1”; *L* is the number of target region values that are considered for the amplification; *k*_4_ has a value of 1 if the exon amplification in question is covered by only one target region, otherwise k_4_ has a value of 0; *E* is the number of exons affected; *Mdist* is the median for the values of the differences between neighboring normalized coverage values.

For LRs with a score higher than the threshold (by default, 9.9 and 2 for the first and second *p*-value evaluation steps, respectively), the *p*-value is calculated by the following bootstrap analysis. When using only the normalized data of samples and target regions covered enough, BRACNAC shuffles the data array randomly by column for each sample and then by rows. The shuffle number of the sample columns is equal to 1000 (by default and can be changed), the shuffle number of the LR target regions (by rows) is equal to the number of samples without potential LRs from the first step (when we use hard CNV and CNA filters, with a minimal score of 9.9 and a maximal *p*-value of 0.01). After each shuffling step, BRACNAC calculates the score similarly as described above but normalizes it only by the length:(3)$$score=\frac{\sum_{i=1}^{N}a_{i}}{1.3^{\frac{M}{L}}}$$
where *a_i_* is a normalized value (if we consider amplifications) or 4.0 minus the normalized value (if we consider deletions); *M* is the total number of normalized values shuffled more than “dupl1” (for amplifications) or less than “del1” (for deletions); *L* is the number of target regions affected by the LR considered. Such a two-step *p*-value evaluation helps us avoid the influence of obviously positive samples on the subsequently calculated *p*-value.

BRACNAC can be applied in command line or graphical user interface (GUI) versions and uses the following Python modules: argparse (to read input arguments), NumPy (for arrays) [17], xlsxwriter (to write outputs into XLS tables), and scipy (for statistics) [18]. The BRACNAC tool is freely available at https://github.com/aakechin/bracnac/ (accessed on 18 November 2023).

### 4.2. Datasets

To compare the BRACNAC performance with those of panelcn.MOPS [6] and MLPA, we used *BRCA1/2*-targeted NGS data previously obtained from 211 ovarian cancer and 2 prostate cancer patient leukocyte DNA samples with an in-house (nine runs) panel and the GeneRead BRCA panel v2 (Qiagen, one run), covering whole-coding regions of the *BRCA1* and *BRCA2* genes [19]. Informed consent was obtained from all patients who participated.

Ten runs were performed using the MiSeq and MiniSeq platforms (Illumina) involving 730 DNA samples from leukocytes and FFPE, from which 213 samples were also selected for MLPA. For these samples, MLPA was performed with the *BRCA1* MRC Holland assay (lot P002-BRCA1-D1-1114) following the manufacturer’s instructions. For 34 of these samples, the *BRCA2* gene (lot P045-BRCA2/CHEK2-B3-0714) was also analyzed. Two or four results were compared for each patient to perform ROC analysis, i.e., *BRCA1* and/or *BRCA2* deletion and/or amplification. For each result compared, the match was considered an event with the same CNV type (deletion or amplification) and target (*BRCA1* or *BRCA2*).

BRACNAC was also tested on targeted NGS of other research groups (NCBI SRA Project PRJNA493651; European Genome-Phenome Archive EGAD00001003400, which was used for panelcn.MOPS) [6,20] and WES (PRJNA388048) data. The first dataset was obtained while discovering germline variants in 192 breast cancer patients using the AmpliSeq *BRCA1* and *BRCA2* panel (Thermo Fisher Scientific) and the Ion Proton platform. The second one was used for testing the panelcn.MOPS program and includes 99 samples tested with the *BRCA1/2* MLPA assay and sequenced with the TruSight^TM^ Cancer (TSC) panel (Illumina). The WES dataset were generated for 60 FFPE DNA samples containing germline *BRCA1/2* mutations, from breast and ovarian cancer patients. However, there was no detailed information about the library preparation protocol or the NGS platform used. To identify short and point mutations, we applied a BRCA analyzer [19]. The WGS data were not used because we could not find such data for enough samples in one NGS run.

### 4.3. Statistics

All plots and statistical tests were performed with Python modules scipy [18], scikit-learn [20], and matplotlib [21]. For ROC analysis, “roc_auc_score” and “roc_curve” functions of the scikit-learn Python module were used.

## 5. Conclusions

In this study we presented a newly developed bioinformatics tool for detecting CNVs and CNAs in the *BRCA1* and *BRCA2* genes using in NGS data. This tool is specifically tailored for data obtained with various library preparation kits and sequencing platforms. Our findings also highlighted the general limitations of such tools, including the minimal number of samples per NGS run and the minimal percent of CNV-/CNA-negative samples. We hope our results will be valuable for researchers and clinicians working in cancer genetics and will contribute to improving the accuracy and sensitivity of CNV and CNA detection in NGS data.

## Figures and Tables

**Figure 1 ijms-24-16630-f001:**
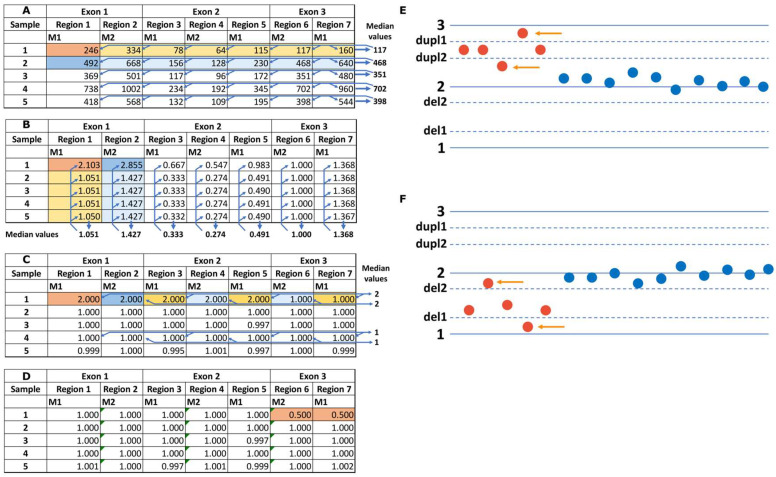
A theoretical example of coverage data processing for calling CNVs and CNAs. In (**A**–**C**), the color-filled cells designate the set of values, the median of which is used to normalize the darker cell. So, each value of the table is divided by the median value of rows (**A**), columns (**B**), or cells of rows corresponding to one multiplex reaction (**C**). In (**D**), the cells of the target regions with one copy deletion are filled in orange. (**A**) Initial coverage values calculated as the median value among all the nucleotides of a target region. The arrows designate the process of the median value calculation and its use for data normalization. (**B**) Coverage values normalized by the median for all sample values. The arrows designate the process of the median value calculation and its use for data normalization. (**C**) Coverage values from step (**B**) normalized by the median for all values of a target region. The arrows designate normalization for the next step and are shown for only two rows. For the other rows, the procedure is the same. (**D**) Coverage values from step (**C**) normalized by the median for the target regions of the same multiplex of the same sample. Finally, these values are multiplied by 2. (**E**,**F**) Examples of “del1”, “del2”, “dupl1”, and “dupl2” thresholds used for calling duplications (**E**) and deletions (**F**). Each dot is the theoretical normalized coverage value calculated as described above and multiplied by 2 for each target region. The horizontal axis represents the number of target regions in the genes studied. For both deletions and amplifications, at least one value should be higher than “dupl1” (for amplifications, indicated by an arrow) or less than “del1” (for deletions, indicated by an arrow), and no more than one value should be less than “dupl2” (for amplifications, indicated by an arrow) or greater than “del2” (for deletions, indicated by an arrow).

**Figure 2 ijms-24-16630-f002:**
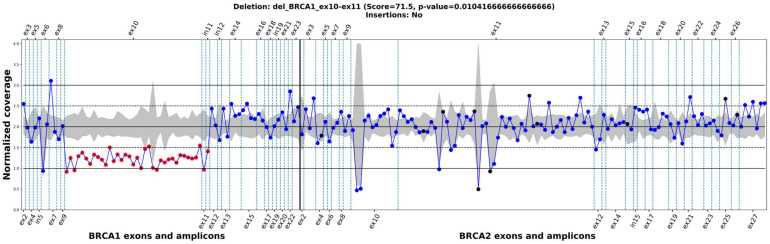
An example of BRACNAC output plot. The blue and red dots designate target regions which were not or were considered as potential CNVs/CNAs, respectively. The black dots are the target regions with coverage values less than the coverage threshold. The final result and the most likely CNV/CNA identified, reported with score value and *p*-value above the plot, are shown. The horizontal axis includes the exon numbers for the *BRCA1* and *BRCA2* genes. The vertical axis reflects the normalized coverage values. The grey area around the line of normalized coverage corresponding to 2 illustrates the first and third quartiles for sample of normalized values for each target region among all samples.

**Figure 3 ijms-24-16630-f003:**
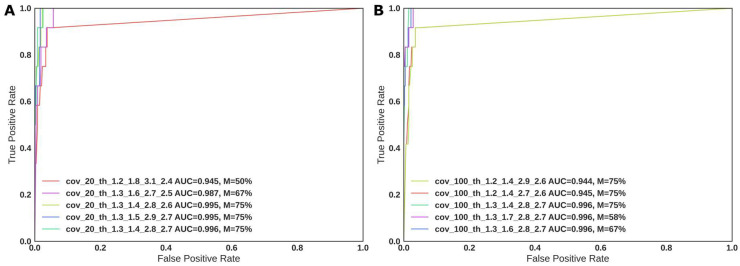
BRACNAC ROC curves for different thresholds, including a minimal median coverage of the sample to be included in the analysis as 20 (**A**) or 100 (**B**). Three curves with the best AUC values and two curves with the worst values are shown to reflect the variation in AUC values. “cov” is the minimal acceptable median coverage depth for a sample and each target region; “th” represents the BRACNAC thresholds for calling CNVs (-del1, -del2, -dupl1, and -dupl2, respectively). AUC is the area under the curve, and M is the percent of CNVs identified as completely matching by the LR boundaries. All combinations of the following values of “del1”, “del2”, “dupl1”, and “dupl2” were tested: 1.2 and 1.3 (“del1”); 1.4, 1.5, 1.6, 1.7, and 1.8 (“del2”); 2.7, 2.8, 2.9, and 3.1 (“dupl1”); 2.4, 2.5, 2.6, and 2.7 (“dupl2”). We found that 12 and 201 samples were CNV-positive or CNV-negative by MLPA, respectively. Thirty-four were also tested for CNVs in the *BRCA2* gene. The deletions and duplications were identified as distinct cases. In total, we considered 494 cases, with 12 identified as CNV-positive, and the others as CNV-negative. For the coverage value of 20, there were 12 and 478 CNV-positive and -negative cases. For the coverage value of 100, there were 12 and 454 CNV-positive and -negative cases.

**Figure 4 ijms-24-16630-f004:**
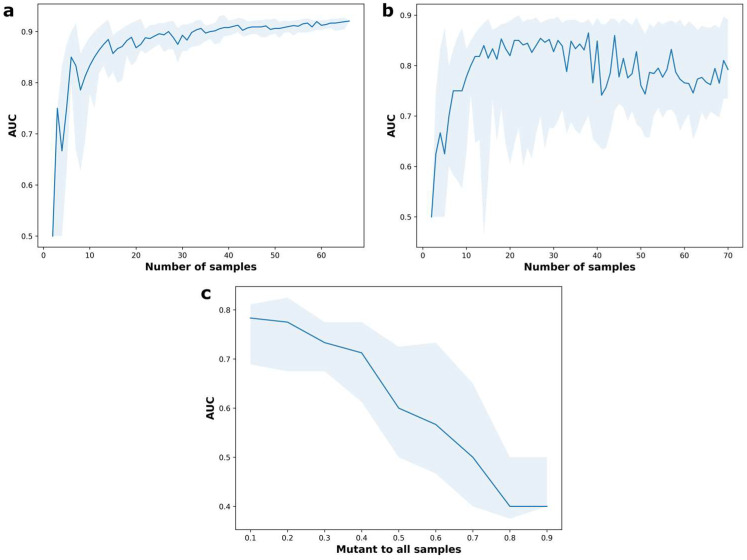
The efficacy of CNV detection for different sets of NGS data samples. The solid line exhibits the mean AUC of CNV detection for 100 random sets of CNV-positive and CNV-negative samples. The blue area around the solid line indicates the first (low boundary) and the third (upper boundary) quartiles of AUC values obtained. (**a**). The dependence of the AUC values on the number of samples in a random NGS dataset used for CNV detection. (**b**). The dependence of the AUC values on the number of samples in a random NGS dataset used for CNV detection allowing for the use of the samples from multiple runs. (**c**). The dependence of the AUC values on the share of CNV-positive samples.

**Figure 5 ijms-24-16630-f005:**
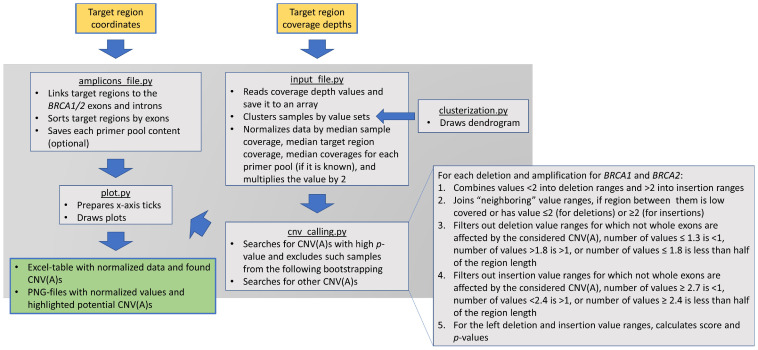
BRACNAC’s main parts and LR calling protocol. Yellow, green, and blue boxes demonstrate input, output, and modules of the program, respectively. The blue arrows designate interactions between the modules of the program. The underlined text represents names of Python-scripts included into the program.

**Table 1 ijms-24-16630-t001:** NGS datasets used in this study.

Step	Type of Dataset	Number of Samples	Method of CNV Validation
- Initial algorithm development - Optimization of threshold values - Comparison with panelcn.MOPS results	in-house targeted NGS data	147	Only *BRCA1* MRC-Holland-179 *BRCA1* and *BRCA2*/*CHEK2* MRC-Holland-34
	GeneRead BRCA panel v2 (Qiagen)	66	
- Comparison with panelcn.MOPS results - Validation using other researchers’ targeted NGS data	TruSight^TM^ Cancer targeted NGS-panel	99	*BRCA1* and *BRCA2* MLPA
- Validation using other researchers’ targeted NGS data	AmpliSeq *BRCA1* and *BRCA2* targeted NGS panel	192	No
- Validation using other researchers’ WES data	Whole-exome sequencing data	60	No

**Table 2 ijms-24-16630-t002:** CNV-positive cases identified with MLPA. The patients with a complete match between the MLPA and the BRACNAC results are shown in bold. del_BRCA1_up-ex2 and del_BRCA1_ex2-ex2 were regarded as matched because the targeted NGS panel used did not cover the *BRCA1* non-coding (exon 1 and exon 2, 5’-ends) and promoter regions.

Sample ID	CNV Detected with MLPA	CNV Detected with BRACNAC
mlpa_1	del_BRCA1_ex20-ex23	**del_BRCA1_ex20-ex23**
**mlpa_2**	**del_BRCA1_ex10-ex11**	**del_BRCA1_ex10-ex11**
**mlpa_3**	**del_BRCA2_ex21-ex24**	**del_BRCA2_ex21-ex24**
**mlpa_4**	**del_BRCA1_up-ex2**	**del_BRCA1_ex2-ex2**
mlpa_5	dupl_BRCA1_ex4-ex6	dupl_BRCA1_ex5-ex6
**mlpa_6**	**del_BRCA1_ex3-ex12**	del_BRCA1_ex4-ex12
**mlpa_7**	**del_BRCA1_ex20-ex21**	**del_BRCA1_ex20-ex21**
**mlpa_8**	**del_BRCA1_up-ex2**	**del_BRCA1_ex2-ex2**
mlpa_9	del_BRCA1_ex19-ex23	**del_BRCA1_ex19-ex23**
mlpa_10	del_BRCA1_ex19-ex23	del_BRCA1_ex20-ex22
**mlpa_11**	**del_BRCA1_up-ex2**	**del_BRCA1_ex2-ex2**
**mlpa_12**	**del_BRCA1_ex20-ex23**	**del_BRCA1_ex20-ex23**

**Table 3 ijms-24-16630-t003:** CNVs detected in the *BRCA1/2* NGS data obtained by Solodskikh et al. [9]. The sample ID corresponds to the NCBI sequence read archive (SRA) accession number. LRs confirmed during the manual figure analysis are in bold.

Sample ID	Age	Short Pathogenic Variants	CNV	*p*-Value
**SRR7910157**	**33**	**No**	**del_BRCA1_ex2-ex22**	**0.001**
SRR7910176	49	No	del_BRCA2_ex2-ex3	0.001
SRR7910204	59	No	dupl_BRCA2_ex3-ex6	0.001
SRR7910262	62	No	dupl_BRCA1_ex21-ex23	0.001
SRR7910265	62	No	del_BRCA1_ex2-ex2	0.001
**SRR7910283**	**35**	**No**	**del_BRCA1_ex23-ex23**	**0.001**

**Table 4 ijms-24-16630-t004:** CNVs/CNAs detected in the PRJNA388048 WES NGS data from FFPE DNA samples. The sample ID corresponds to the NCBI SRA accession number. Likely true-positive LRs are highlighted in bold.

Sample ID	Age	Cancer	Short Pathogenic Variants	CNV	*p*-Value
**SRR5604273**	**55**	**Ovary**	**No**	**del_BRCA1_ex13-ex13**	**0.018**
**SRR5604275**	**33**	**Breast**	***BRCA2* c.271_271delTA**	**del_BRCA1_ex2-ex23**	**0.001**
**SRR5604279**	**58**	**Breast**	***BRCA1* c.5266dupC**	**del_BRCA1_ex17-ex21**	**0.018**
**SRR5604281**	**43**	**Breast**	***BRCA2* c.5946delT**	**dupl_BRCA1_ex13-ex23**	**0.001**
**SRR5604292**	**35**	**Breast**	***BRCA2* c.8364G>A**	**dupl_BRCA1_ex2-ex23**	**0.001**
**SRR5604295**	**39**	**Breast**	**No**	**del_BRCA1_ex13-ex19**	**0.001**
**SRR5604298**	**57**	**Ovary**	***BRCA1* c.68_69delAG**	**del_BRCA1_ex11-ex12**	**0.001**
**SRR5604299**	**49**	**Breast**	***BRCA2* c.5645C>A**	**del_BRCA1_ex2-ex23**	**0.018**
SRR5604308	43	Breast	*BRCA2* c.3922G>T	dupl_BRCA1_ex17-ex22	0.001
SRR5604312	34	Breast	*BRCA1* c.68_69delAG	del_BRCA2_ex15-ex18	0.001
SRR5604313	50	Breast	*BRCA2* c.1054dupT	dupl_BRCA1_ex15-ex16	0.001
**SRR5604314**	**40**	**Ovary**	***BRCA1* c.3155delA**	**dupl_BRCA1_ex13-ex17**	**0.001**
				**dupl_BRCA1_ex19-ex23**	**0.001**
SRR5604315	40	Ovary	*BRCA1* c.3155delA	del_BRCA1_ex12-ex13	0.001

## Data Availability

The BRACNAC tool is freely available at: https://github.com/aakechin/bracnac/ (accessed on 18 November 2023). All data analyzed are in the Appendix A.

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
