# Peer review of "BRACNAC: A BRCA1 and BRCA2 Copy Number Alteration Caller from Next-Generation Sequencing Data"

_ijms, 2023, doi:10.3390/ijms242316630_

Round 1
Reviewer 1 Report
Comments and Suggestions for Authors
I thank the Academic Editor for giving me the opportunity to review this manuscript entitled 'BRACNAV: BRCA1 and BRCA2 Copy Number Alterations Viewer from next-generation sequencing data'. I believe the authors' intent is commendable as it is important to conduct quality research in newly explored territories such as Next Generation Sequencing (NGS) applied to gene mutations such as BRCA1/2. In an attempt to do this, the authors started from a large cohort (more than 200) of ovarian and prostate carcinoma samples, and processed a library of NGS data, with particular copy number variations and alterations in these genes.
Congratulations to the authors and I commend the manuscript for publication in the prestigious journal IJMS. I recommend a minor revision for the following reasons:
For the first time, it is always better to give the full name and after the abbreviation.
Comments on the Quality of English LanguageModerate editing.
Author Response
First of all, we thank all reviewers for their suggestions on the improvements for our manuscript. Below, we provide answers to all questions and suggestions.
- For the first time, it is always better to give the full name and after the abbreviation.
We provided full name for MLPA, written without full name in the Abstract.
- Moderate editing of English language required
We corrected English grammar mistakes.
Reviewer 2 Report
Comments and Suggestions for Authors
Authors in the manuscript “BRACNAV: BRCA1 and BRCA2 Copy Number Alterations
Viewer from next-generation sequencing data” present an algorithm to call BRCA1 and BRCA2 variants. However, the title communicated that this is a viewer, not algorithm , therefore it is a bit confusing, since by viewer people expect a software tool like Integrative Genome Viewer (IGV) or similar. It is not a viewer, but a CNV calling algorithm that is described.
As such, the manuscript is lacking more background such as existing algorithms in the field that are extensively used for BRCA1 and BRCA2 variants detection. Also, in the Methods description of the algorithm a rationale for choosing those specific values of the set thresholds needs to be provided. For example why 1.3 and 1.8 and not 1.4 and 1.9 ? Please provide details why and how these thresholds were chosen. If they were recommended by literature, then there must be references. If they were found in numerical experiment, then the experiments performed need to be mentioned and described.
In Results it would help a reader to have a table with the clear indication of the numbers of total samples and how many samples were used in each experiment actually and the data sources. This information is conveyed in a confusing way, many details obscuring this information. Similarly a description of datasets in 4.2.
Other- the ROC curves in Figure 1 demand better explanation. First, it would be more informative to have two Figures one for coverage 20 and another for coverage 100 and the threshold values shown on the plot next to the curve; Second , please explain – provide a formula- how you calculated True positive and False positive rates using the positive and negative samples and how many positive and negative samples there were.
Other that that – it is a useful manuscript and experiments and algorithm.
Comments on the Quality of English Language
Grammar : 4.1 “parts for effectrive calling CNVs” .
Author Response
First of all, we thank all reviewers for their suggestions on the improvements for our manuscript. Below, we provide answers to all questions and suggestions.
- However, the title communicated that this is a viewer, not algorithm , therefore it is a bit confusing, since by viewer people expect a software tool like Integrative Genome Viewer (IGV) or similar. It is not a viewer, but a CNV calling algorithm that is described.
We replaced BRACNAV with BRACNAC (caller) in the program’s name, code and the manuscript’s title.
- As such, the manuscript is lacking more background such as existing algorithms in the field that are extensively used for BRCA1 and BRCA2 variants detection.
We added more information about existing tools in the Introduction section.
- Also, in the Methods description of the algorithm a rationale for choosing those specific values of the set thresholds needs to be provided. For example why 1.3 and 1.8 and not 1.4 and 1.9 ? Please provide details why and how these thresholds were chosen. If they were recommended by literature, then there must be references. If they were found in numerical experiment, then the experiments performed need to be mentioned and described.
Thanks for this suggestion. These values were initially chosen from logic that we described in the new subsection 2.1 “Algorithm” of the Results. Also, we added new Figure 2 that describes the process of data normalization and use of these thresholds.
- In Results it would help a reader to have a table with the clear indication of the numbers of total samples and how many samples were used in each experiment actually and the data sources. This information is conveyed in a confusing way, many details obscuring this information. Similarly a description of datasets in 4.2.
We added Table 1 with information about all datasets used in the study.
- Other- the ROC curves in Figure 1 demand better explanation. First, it would be more informative to have two Figures one for coverage 20 and another for coverage 100 and the threshold values shown on the plot next to the curve; Second , please explain – provide a formula- how you calculated True positive and False positive rates using the positive and negative samples and how many positive and negative samples there were.
We divided ROC-curves onto two figures: for coverage depth values 20 and 100 (now, it is Figure 2). In the Methods sections we added functions that were used for ROC-analysis:
“For ROC-analysis, “roc_auc_score” and “roc_curve” functions of scikit-learn Python modules were used.”
Also, in the caption of Figure 2, we added the number of positive and negative samples, included into analysis:
“12 and 202 samples were CNV-positive or CNV-negative by MLPA, respectively. Thirty-four were also tested for CNVs of the BRCA2 gene. The deletions and duplications were identified as distinct cases. In total, we considered 496 cases with 12 identified as CNV-positive, and others as negative. For the coverage value of 20, there were 12 and 478 CNV-positive and negative cases. For the coverage value of 100, there were 12 and 454 CNV-positive and negative cases.”
However, we think that it will not be more clearly if we show threshold values next to the curves. In the current variant, most of the curves are almost identical, and only several curves differ from other, and in our opinion, they can be clearly identified from other by color.
- Grammar : 4.1 “parts for effectrive calling CNVs”
We corrected this mistake.
Reviewer 3 Report
Comments and Suggestions for Authors
Kechin et al. present a program for BRCA copy number alteration analysis. However, it is not really clear what the authors want. Is it a viewer, as the title suggests? Then the paper should contain screenshots and information on customizing the view. Or is it a tool for *detection* of alterations? Then the performance should be analyzed in greater depth and the title should be changed. The authors present an overwhelming mass of 40 (!) supplementary figures which basically all show the same kind of plot. This must be aggregated to a manageable number of figures. If MLPA is the gold standard, what is the advantage of your method then? Why should I use it?
p. 1 Abstract: explain the term "MLPA"
p. 2 Introduction: include reference on MLPA and panelcln.MOPS, explain how it works. The introduction must also include and discuss the basic idea of the new algorithm.
p. 2 Results: Here, the new algorithm must be explained in detail before any results are presented. The methods section is just for technical details, not for the basics of the algorithm.
Supplementary figures: 40 are way too many. Show and explain one of these figure triples in the paper itself and reduce the supplement to <=10 figures at most. MLPA, BRACNAV and panelcln.MOPS must be scaled to the same x-axis to be comparable in all figures and plotted directly on top of each other. The result of each of the three sets of supplementary figures can be condensed to three figures showing, as lines, just the detected CNAs per algorithm in different colors (maybe similar to the panelcln.MOPS plots). Only one example of the current supplementary figures should be shown in the manuscript itself.
Fig. 1: It does not make sense to show alterations of the parametrs of your program unless you explain them. At this point, the reader cannot understand all these parameters.
p. 2 Results: "there were four [...] or two comparison points" What are "comparison points? I do not understand what is meant by that.
The program often outputs a p-value of 0. That should not occur, calculate a meaningful small p-value instead.
Where did you get the NGS data to analyze from? That is not described well in the paper. Present the papetrs where the original data comes from. It must be clear for each section where the data comes from.
p. 3 "Table 2 and supplementary figures S1-S12" probably S13-S21
p. 6 M&M "It uses the target region coordinates file (maybe the default)" what does "maybe the default" mean? How does the coordinate file look like? Present an example. Same for the TSV table of coverages; explain the format.
p. 6 M&M: "normalized by the following types: [...]" why do you use four different normalizations?? Explain the normalizations, Nr. 2-4 are not comprehensible as presented now.
p. 7 The method is not comprehensible at all. Describe oit with a figure showing an artificial example what exactly you do, especially how you combine four different normalizations.
p. 6. 7 equations: How did you make up these equations? What is the rational behind them? 1/L, 1/E or k4 in the exponent, numbers 1.3 and 4 all seem very arbitrary. The denominator should better be written as 1.3^(1/L+k4*2+1/E+Mdist). The whole calculation of the score seems very weird.
Author Response
First of all, we thank all reviewers for their suggestions on the improvements for our manuscript. Below, we provide answers to all questions and suggestions.
- However, it is not really clear what the authors want. Is it a viewer, as the title suggests? Then the paper should contain screenshots and information on customizing the view. Or is it a tool for *detection* of alterations? Then the performance should be analyzed in greater depth and the title should be changed.
We replaced BRACNAV with BRACNAC (caller) in the program’s name and codes and the manuscript’s title.
- The authors present an overwhelming mass of 40 (!) supplementary figures which basically all show the same kind of plot. This must be aggregated to a manageable number of figures.
In this particular case, we allow ourselves to disagree with the reviewer’s suggestion because all figures are in the Supplementary materials, and, in our opinion, they do not overload the manuscript’s main text. However, all these figures help readers, reviewers, and potential program users to look at real results of data processing. There are many tools described for CNV detection and NGS data analysis. However, it is an often case when a described bioinformatic program does not produce results described in the article, is unavailable (e.g., PattRec or smCounter2 tools) or figures produced by the program were interpreted subjectively. Therefore, we tried to provide figures in the native format as they had been produced by the program to give the reader opportunity to interpret them looking at all values with which program accompanies the result. We think it should be the standard of data representation for new bioinformatic tools like whole western-blot images for corresponding studies.
- If MLPA is the gold standard, what is the advantage of your method then? Why should I use it?
Targeted NGS is a commonly used approach for detection of BRCA1 and BRCA2 short mutations (single-nucleotide variants, short insertions, or deletions). However, due to absence of any gold-standard bioinformatics tool for identification of CNVs and CNAs in targeted NGS data, especially for BRCA1 and BRCA2 genes, their identification is carried out with additional molecular methods, e.g., MLPA or ddPCR. The described BRACNAC tool can be used with both targeted NGS or whole-exome sequencing data, simplifying the process of CNV and CNA detections, because additional molecular tests can be replaced with bioinformatics data analysis. We added the following text into the Introduction:
“…and/or multiplex ligation-dependent probe amplification (MLPA) [4]. The latter has long been considered the gold-standard approach for CNV detection. The methodology in-volves the amplification of probes that can be ligated if only both were hybridized to the neighboring regions, yielding PCR products with different lengths for multiple target regions [5]. It would be more efficient to detect CNVs and CNAs along with short mu-tations routinely identified with targeted NGS. Several new algorithms of CNV detec-tion were developed for targeted NGS data, with some organized into bioinformatic tools: panelcn.MOPS [6], CNVpytor [7], or several simple R scripts [8].”
- p. 1 Abstract: explain the term "MLPA"
We provided full name for MLPA, written without full name in the Abstract.
- p. 2 Introduction: include reference on MLPA and panelcln.MOPS, explain how it works. The introduction must also include and discuss the basic idea of the new algorithm.
We added information about MLPA, panelcn.MOPS, and BRACNAC algorithms in the Introduction:
“The methodology involves the amplification of probes that can be ligated if only both were hybridized to the neighboring regions, yielding PCR products with different lengths for multiple target regions [5].”
“For example, R-scripts developed by Singh et al. [8] create a pool of normal samples, evaluate the coverage for many sliding windows, and calculate the ratio of coverage depth for the query sample and pool of normal samples for each nucleotide of sliding windows. Panelcn.MOPS additionally applies one-step normalization to the third quartile, also selecting control samples.”
“The method involves several steps of data normalization and identification of sets of target regions with elevated or lowered copy numbers. Of critical relevance is the esti-mation of the probability of an observed copy number increase or decrease being ren-dom, i.e., the p-value.”
- p. 2 Results: Here, the new algorithm must be explained in detail before any results are presented. The methods section is just for technical details, not for the basics of the algorithm.
Thanks for this suggestion. We explained algorithm including threshold values in the 2.1 subsection of the Results.
- Supplementary figures: 40 are way too many. Show and explain one of these figure triples in the paper itself and reduce the supplement to <=10 figures at most. MLPA, BRACNAV and panelcln.MOPS must be scaled to the same x-axis to be comparable in all figures and plotted directly on top of each other. The result of each of the three sets of supplementary figures can be condensed to three figures showing, as lines, just the detected CNAs per algorithm in different colors (maybe similar to the panelcln.MOPS plots). Only one example of the current supplementary figures should be shown in the manuscript itself.
We understand that this number of supplementary figures may be redundant. However, we tried to provide readers with raw materials produced by the programs tested. Of course, we could draw new figures based on the programs’ results skipping its variation. However, then the reader will not see the variability of results which may lead to false positive or false negative results. For example, Figure S1 shows real Figure of the MLPA result (we can see that one MLPA amplicon is near the MLPA threshold for deletions); for BRACNAC, there are many amplicons having higher or lower normalized coverage values but they were not interpreted as false-positive CNVs; for panelcn.MOPS, we can see many dots which represent amplifications or deletions which users should interpret by themselves. And in our opinion, it is useful to look at all these figures to understand variation of results which some program possesses. And it is really hard to evaluate some new programs which do not provide such figures, especially when new programs are not available after the manuscript’s publication as we wrote it above. Therefore, unfortunately, we cannot make single scale for MLPA, BRACNAC and panelcln.MOPS figures.
- Fig. 1: It does not make sense to show alterations of the parametrs of your program unless you explain them. At this point, the reader cannot understand all these parameters.
We explained algorithm including threshold values in the 2.1 subsection of the Results and replaced Figure 1 (now it is Figure 2) according to another reviewer’s suggestion. Also, we added more details into the Figure’s caption.
- p. 2 Results: "there were four [...] or two comparison points" What are "comparison points? I do not understand what is meant by that.
We replaced “comparison points” with “comparisons”. Also, in the Materials and Methods, the following details were described:
“Two or four results were compared for each patient to perform ROC-analysis: BRCA1 and/or BRCA2 deletion and/or amplification.”
For example, if some sample was positive for deletion of some BRCA1 exons, we have the following true results:
- BRCA1 deletion positive
- BRCA1 amplification negative
- BRCA2 deletion negative
- BRCA2 amplification negative
And all these four results were considered as distinct four cases. Therefore, total number of cases were the number of samples multiplied by four.
- The program often outputs a p-value of 0. That should not occur, calculate a meaningful small p-value instead.
Thanks for this suggestion. We started bootstrap analysis from one to avoid cases with p-value of 0. This led to increase in the AUC values obtained. We replaced all p-values in all tables and figures.
- Where did you get the NGS data to analyze from? That is not described well in the paper. Present the papetrs where the original data comes from. It must be clear for each section where the data comes from.
We added Table 1 in the Results section. It includes information about all datasets used. Also, all information about these datasets is described in the Materials and Methods section (subsection 4.2. Datasets)
- p. 3 "Table 2 and supplementary figures S1-S12" probably S13-S21
We corrected it. Now Table 2 is Table 3, and there are supplementary figures S13-S18 instead of S13-S21.
- p. 6 M&M "It uses the target region coordinates file (maybe the default)" what does "maybe the default" mean? How does the coordinate file look like? Present an example. Same for the TSV table of coverages; explain the format.
We added an example file with target region coordinates and example input file into the GitHub repository. It is based on coordinates of our in-house NGS-panel. We removed the confusing phrase “maybe the default”.
- p. 6 M&M: "normalized by the following types: [...]" why do you use four different normalizations?? Explain the normalizations, Nr. 2-4 are not comprehensible as presented now.
We added explanations about the normalization steps in the Results section (subsection 2.1 Algorithm).
- p. 7 The method is not comprehensible at all. Describe oit with a figure showing an artificial example what exactly you do, especially how you combine four different normalizations.
We added Figure 1 with an example of data normalization steps.
- p. 6. 7 equations: How did you make up these equations? What is the rational behind them? 1/L, 1/E or k4 in the exponent, numbers 1.3 and 4 all seem very arbitrary. The denominator should better be written as 1.3^(1/L+k4*2+1/E+Mdist). The whole calculation of the score seems very weird.
We added logic of threshold value selection and equations’ structure in the new subsection 2.1 Algorithm. Also, we corrected the denominator as suggested.
Round 2
Reviewer 3 Report
Comments and Suggestions for Authors
The authors present a revised version of their manuscript on a BRCA CNV caller. However, the method is still not explained in a comprehensible way and the equations used seem very arbitrary. The english language is very hard to understand and should be corrected by a native speaker.
The manuscript does not contain line numbers, which makes the review process very cumbersome.
p. 2, results section: "algorithm and initial threshold values were chosen..." At this point, the reader does not know what is meant by "threshold values". You should explain the algorithm first.
Tab. 1 the refeences for the data sets not generated in house should be specified. Which data exactly did you use, where does it come from (if not in house)
p. 3: all these normalizations are not comprehensible. Explain in detail what the input data is comprised of. Is it a set of coverages per nucleotide? Obviously not, Fig. 1 shows numbers (whatever they might mean) per "region" of an exon. Where do these regions come from? What are their boundaries? What do the numbers in Fig. 1 mean? Is it some sort of coverage? Mean? Median? In the text, explain comprehensibly which normalizations are performed row-wise, which column-wise. I guess that A is the original data and B, C, D are first, second, third normalization, but thatz should be clarified. If the first normalization is between samples, as stated in the text, I would expect each *column* of Fig. 1B having values below and above 1; however the first and second column all have values above 1, the third column all have values below one. Accordingly, if the second normalization is done between regions (as stated in the text), I would expect all *rows* to have numbers above and below 1 in Fig. 1C. That is again not the case. Why are the numbers in Fig. 1C mostly 1.000 and 2.000? a normalization of the values in Fibg. 1B should lead to crooked numbers. Are the normalizatiopns done by dividing or subtracting? Is the mean of the median value used?
still p. 3: "these deviating values follow each other, forming continuous set" -- what does that mean?
p. 3: "may demonstrate the values lower deviation than one copy" -- what does that mean?
p. 3: the initial algorithm involved determining four thresholds" -- how were the thresholds chosen? Based on sensitivity/specificity of detection? If so, what were the limits of sensitivity/specificity at which the thresholds were set?
p. 3: "The second values [...] are used for calculating the CNV/CNA score" -- If they are "thresholds" that would mean that you predict all values obeying the thresholds as CNV, without the need for a scoring function. Why do you need a scoring function if you already have thresholds for a "yes/no" decision? A threshold is a threshold and not a parameter in a scoring function.
Fig. 1: "color filled cells designate trhe set of values, the median of which is used to normalize darker cell" -- not comprehensible.
Fig. 1: E and F are examples of..." What exactly does the figure show? What are the bullets? What does their position mean? Why are there arrows in the figure? What does the color of the bullets mean?
How many of the regions (or positions?) lie within the threshold, how many outside? What happens to the ones outside, are they discarded? Now that you have thresholds, your calling is finished (within threshold: yes, outside threshold: no), what is the scoring good for?
p. 4: "By combining neighboring target regions" -- how are they "combined"? How many and which "neighboring regions" are aggregated?
p. 4 "we collect different overlapping sets [...] and the current target regions among all the samples" -- not comprehensible.
p. 5: Up to now you have tzhreshold values which cvan only be used to classify as "yes" or "no". There is no scoring described in the results section. With a binary classification, you cannot plot ROC curves or calculate AUC values.
p. 10 score equation: That equation looks completely arbitrary. Why do you put all that stuff in an exoponent of 1.3? why are all the things in the exponent summed up? "k1 is one if all consiudered exons ..." the complete paragraph is not comprehensible.what is. e.g., the "median of differences between all neighboring values" or "the target region values that are no more than 1.7"? Are all those nombers in the range of 0 and 1?
p. 11: "for the first and second CNV calling steps" you did not mention two different "steps" before. Do both or just one score have to be above the threshold?
At least one of the supplementary figures must be included in the manuscript itself to show an example of the output of the algorithm.
Especially when the authors try to explain their method and the different normalizations, their language is barely comprehensible.
Author Response
We are grateful to the reviewers for their valuable contributions in improving the manuscript. We tried to take all the comments into account.
- However, the method is still not explained in a comprehensible way and the equations used seem very arbitrary. The english language is very hard to understand and should be corrected by a native speaker.
We have substantially improved the method description, added one new figure (Figure 2), and improved the description of several figures (details below). Also, we have made revisions concerning the English language. To improve the comprehension of the algorithm description, we have replaced all the values related to the normalized coverage values written in words (e.g., one, two, three, four) with the float values (e.g., 1.0, 2.0, 3.0, and 4.0) (lines 107–116, 150, 331–333, 349–370, and 381). The same was done for k1–k4 constants, with the words replaced with integer values (“1” instead of “one”).
- The manuscript does not contain line numbers, which makes the review process very cumbersome.
We have added line numbers. We regret they were lost during the transfer process to the Journal’s template.
- p. 2, results section: “algorithm and initial threshold values were chosen...” At this point, the reader does not know what is meant by “threshold values”. You should explain the algorithm first.
We have replaced this phrase with “algorithm and initial values of different parameters were chosen…” (lines 83–84).
- Tab. 1 the references for the data sets not generated in house should be specified. Which data exactly did you use, where does it come from (if not in house)
The references for these datasets are listed in the Materials and Methods section. We decided not to provide references in Table 1 so as not to overload it.
- p. 3: all these normalizations are not comprehensible. Explain in detail what the input data is comprised of. Is it a set of coverages per nucleotide? Obviously not, Fig. 1 shows numbers (whatever they might mean) per “region” of an exon. Where do these regions come from? What are their boundaries? What do the numbers in Fig. 1 mean? Is it some sort of coverage? Mean? Median? In the text, explain comprehensibly which normalizations are performed row-wise, which column-wise. I guess that A is the original data and B, C, D are first, second, third normalization, but thatz should be clarified. If the first normalization is between samples, as stated in the text, I would expect each *column* of Fig. 1B having values below and above 1; however the first and second column all have values above 1, the third column all have values below one. Accordingly, if the second normalization is done between regions (as stated in the text), I would expect all *rows* to have numbers above and below 1 in Fig. 1C. That is again not the case. Why are the numbers in Fig. 1C mostly 1.000 and 2.000? a normalization of the values in Fibg. 1B should lead to crooked numbers.
We have added additional information to Figure 1, with the arrows showing the normalization steps. Also, we have provided a more elaborate caption for Figure 1. We assume that the misunderstanding of the process of getting the numbers was due to the insufficient description of the normalization process. We have tried to correct this and hope that the current description is now sufficiently detailed (lines 128–147).
“Figure 1. A theoretical example of coverage data processing for calling CNVs and CNAs. For A, B, and C, the color-filled cells designate the set of values, the median of which is used to normalize the darker cell. So, each value of the table is divided by the median value of rows (A), columns (B), or cells of rows corresponding to one multiplex reaction (C). For D, the cells of target regions with one copy deletion are filled in orange. A illustrates the initial coverage values calculated as the median value among all the nucleotides of a target region. The arrows designate the process of median value calculation and its use for data normalization. B represents the coverage values normalized by the median for all sample values. The arrows designate the process of median value calculation and its use for data normalization. C reveals the coverage values from step B normalized by the median for all values of a target region values. The arrows designate normalization for the next step and are shown for only two rows. For other rows, the procedure is the same. D demonstrates the coverage values from step C normalized by the median for the target regions of the same pool multiplex of the same sample. Finally, these values are multiplied by 2. E and F are the examples of “del1”, “del2”, “dupl1”, and “dupl2” thresholds used for calling duplications (E) and deletions (F). Each dot is the theoretical normalized coverage value calculated as described above and multiplied by 2 for each target region. The horizontal axis represents the number of target regions of the genes studied. For both deletions and amplifications, at least one value should be more than “dupl1” (for amplifications, indicated by an arrow) or less than “del1” (for deletions, indicated by an arrow), and no more than one value should be less than “dupl2” (for amplifications, indicated by an arrow) or more than “del2” (for deletions, indicated by an arrow).”
Also, we have added new information about the coverage values in the Materials and Methods section (lines 317–320):
“The coverage values for each target region can be mean, median, minimal, or maximal coverage values per all nucleotides of the region, but they should be uniform across all the samples analyzed. In the case of amplicon-based targeted NGS panels, such regions can be represented by the genomic coordinates of the amplified regions.”
- Are the normalizations done by dividing or subtracting? Is the mean of the median value used?
The normalization is done by dividing by the median value. We have added this information to the Figure 1 caption and to the Algorithm description (lines 105–106):
“The normalization steps are carried out by dividing each value by the median value of the corresponding sets of values”.
- still p. 3: “these deviating values follow each other, forming continuous set” -- what does that mean?
We have added the indefinite article “a” that was missing (line 112). This phrase means that if values deviate from 2.0 randomly, they are more likely not to form continuous sets of values, but they will be randomly scattered among all the values.
- p. 3: “may demonstrate the values lower deviation than one copy” -- what does that mean?
We have added the missing preposition “with” (line 114). This phrase means that some values included in the sets of potential deletions or duplications may have a lower deviation from two copies, e.g., 1.3 instead of 1.0 (as expected for deletions) or 2.7 instead of 3.0 (as expected for duplications).
- p. 3: the initial algorithm involved determining four thresholds” -- how were the thresholds chosen? Based on sensitivity/specificity of detection? If so, what were the limits of sensitivity/specificity at which the thresholds were set?
The thresholds were chosen based on the logic described at the beginning of the Results section (subsection 2.1 Algorithm). This logic is similar to the one used in the MLPA approach (see Supplementary Figures S1–S12), where the program searches for the values lower (for deletions) or higher than the thresholds shown in Figures S1–S12.
- p. 3: “The second values [...] are used for calculating the CNV/CNA score” -- If they are “thresholds” that would mean that you predict all values obeying the thresholds as CNV, without the need for a scoring function. Why do you need a scoring function if you already have thresholds for a “yes/no” decision? A threshold is a threshold and not a parameter in a scoring function.
A yes/no decision is initially made for distinct values of each target region. However, we need to identify the exons that are deleted or amplified. Therefore, the conversion of these yes/no decision sets into the decisions for the studied exons necessitates a scoring function to evaluate the probabilities for the changes in copy numbers observed to be random. Thus, the scoring function helps us move from the values of distinct target regions to the values characterizing the changes in the copy numbers of exons.
- Fig. 1: “color filled cells designate trhe set of values, the median of which is used to normalize darker cell” -- not comprehensible.
We have added the arrows indicating the process of calculating median values and their subsequent use for data normalization.
- Fig. 1: E and F are examples of...” What exactly does the figure show? What are the bullets? What does their position mean? Why are there arrows in the figure? What does the color of the bullets mean?
We have added the following additional description in the Figure’s caption (lines 139–147):
“E and F are examples of “del1”, “del2”, “dupl1”, and “dupl2” thresholds used for calling duplications (E) and deletions (F). Each dot is the theoretical normalized coverage value calculated as described above and multiplied by 2 for each target region. The horizontal axis represents the number of target regions of the genes studied. For both deletions and amplifications, at least one value should be more than “dupl1” (for amplifications, pointed with the arrow) or less than “del1” (for deletions, pointed with the arrow), and no more than one value should be less than “dupl2” (for amplifications, pointed with the arrow) or more than “del2” (for deletions, pointed with the arrow).”
- How many of the regions (or positions?) lie within the threshold, how many outside? What happens to the ones outside, are they discarded? Now that you have thresholds, your calling is finished (within threshold: yes, outside threshold: no), what is the scoring good for?
If a sample does not contain any deletions or amplifications, all values should be near 2.0, as commonly observed. For samples containing possible deletions or amplifications, we consider only the sets of target regions that include at least one value that is more than “dupl1” (for amplifications) or less than “del1” (for deletions) and no more than one value less than “dupl2” (for amplifications) or more than “del2” (for deletions). The identification of deletions or amplifications of values can be done through an unlimited number of target regions. However, for copy number variations that encompass a greater number of target regions, the evaluated score will be higher.
If we have understood correctly, the reviewer used the “outside threshold” to identify the normalized coverage values less than “dupl2” (for amplifications) or more than “del2” (for deletions). In these cases, consecutive values outside the thresholds are not considered potential CNV/CNA. However, suppose such a value is located between two values within the threshold. In that case, we include it in the potential CNV (e.g., one of the target regions of BRCA1 exon 10 in Supplementary Figure S6 and the target region of exon 20 in Supplementary Figure S9).
The scoring function allows us to exclude possible false-positive CNVs/CNAs that can be called due to natural variations in the coverage of each target region. Most of the exons studied are covered by several target regions (e.g., PCR amplicons or hybridization oligonucleotides). Thus, when a series of several neighboring target regions exhibits normalized coverage values within the threshold, the probability of a false-positive result decreases, resulting in an increase in the scoring function value. These cases are illustrated in the Supplementary Figures with the BRACNAC plots.
- p. 4: “By combining neighboring target regions” -- how are they “combined”? How many and which “neighboring regions” are aggregated?
We have paraphrased this sentence in the following way:
“We collect different overlapping sets of target regions by extracting the normalized coverage values of neighboring target regions exhibiting copy numbers below 2.0 (for deletions) or above 2.0 (for amplifications). Subsequently, we calculate the scores and estimate the p-value based on the probability of a given set of target regions having such a score at random.”
For example, Supplementary Figure S7 demonstrates two values lower than “del2” (they are about 1.2), for BRCA1 exons 20 and 21. These two values are combined into one continuous set of values and are considered a potential deletion: [1.159, 1.248]. Supplementary Figure S9 shows possible continuous sets of values that can be considered a potential deletion in the BRCA1 gene:
- from exon 19 to exon 23: [1.25, 0.8, 1.75, 1.2, 1.25, 1.2, 1.1]
- from intron 19 to exon 23: [0.8, 1.75, 1.2, 1.25, 1.2, 1.1]
- from exon 20 to exon 23: [1.75, 1.2, 1.25, 1.2, 1.1]
- from exon 21 to exon 23: [1.2, 1.25, 1.2, 1.1]
- from exon 22 to exon 23: [1.25, 1.2, 1.1]
- from exon 19 to exon 22: [1.25, 0.8, 1.75, 1.2, 1.25] and so on…
Then, we should choose what set of values is less likely to be false positive. So, we calculate score values for each of these sets of values and then evaluate the p-value. CNV with the lowest p-value is written to output.
- p. 4 “we collect different overlapping sets [...] and the current target regions among all the samples” -- not comprehensible.
The overlapping sets are collected for one sample, as described in the answer to question #14. However, all overlapping sets have their own score value. For the example described above, we can calculate the scores (here, they are calculated as a simple sum of (4-value) to omit the details):
- from exon 19 to exon 23: [1.25, 0.8, 1.75, 1.2, 1.25, 1.2, 1.1], the score is 19.45
- from intron 19 to exon 23: [0.8, 1.75, 1.2, 1.25, 1.2, 1.1], the score is 7
- from exon 20 to exon 23: [1.75, 1.2, 1.25, 1.2, 1.1], the score is 5
- from exon 21 to exon 23: [1.2, 1.25, 1.2, 1.1], the score is 25
- from exon 22 to exon 23: [1.25, 1.2, 1.1], the score is 45
- from exon 19 to exon 22: [1.25, 0.8, 1.75, 1.2, 1.25], the score is 13.75
Thus, it can be seen that potential CNVs with a higher number of values have higher score values. However, some values included in the sets may be so low because many other samples exhibit values that are similar or even lower for the same target regions but are distributed randomly among all values. Such cases should be excluded from potential deletions. For example, let’s consider the BRCA1 exon 11 in Supplementary Figure S11. The target region of this exon has a value of about 1.2. However, the high variability of this target region coverage among all samples (it is designated by a grey color around blue points and lines) causes this deletion to be discarded. However, we see that panelcn.MOPS did not discard it. Therefore, we evaluate the p-value following the algorithm described (lines 148–160).
- p. 5: Up to now you have tzhreshold values which cvan only be used to classify as “yes” or “no”. There is no scoring described in the results section. With a binary classification, you cannot plot ROC curves or calculate AUC values.
All thresholds, scores, and evaluations of p-value are described in the Materials and Methods section. We believe it is redundant to repeat it in the Results section. We calculate the p-value used for ROC analysis for each potential deletion or duplication and calculate the AUC values. Thus, it is not a binary classification that BRACNAC provides as output. Instead, it detects CNVs/CNAs and calculates the p-value for each prediction.
- p. 10 score equation: That equation looks completely arbitrary. Why do you put all that stuff in an exoponent of 1.3? why are all the things in the exponent summed up? “k1 is one if all consiudered exons ...” the complete paragraph is not comprehensible.what is. e.g., the “median of differences between all neighboring values” or “the target region values that are no more than 1.7”? Are all those nombers in the range of 0 and 1?
We recognize that the equation can appear completely arbitrary. However, almost all the equations ever constructed to predict or identify any biological properties or values appear arbitrary. Machine learning and neural networks represent the most extreme manifestations of this phenomenon. The modus operandi and dependencies of their mechanisms are unknown, yet they are universally applied. This equation is based on natural learning. Furthermore, we have substantiated its effectiveness on a wide range of samples, surpassing 213 in total. This equation was formulated based on the following set of facts:
- A number of normalized values more than 1.3 (or “del1”) increases the probability for this deletion to be false positive. This is why “D/L” is in the exponent.
- If the exon is covered by only one target region, the probability for the CNV of this single exon to be false positive is higher. This is why k4 is in the exponent. And it is multiplied by two to increase this dependence.
- For a higher number of exons deleted, the probability for this observation to be random is higher. This is why 1/E is in the exponent.
- If the normalized coverage values exhibit a high variability, the probability for this CNV/CNA to be false positive is higher. To estimate this variability, we calculate Mdist. We paraphrased its definition in the following way: “Mdist is the median value calculated from the differences between neighboring normalized coverage values”. The equation is as follows:
We incorporated all these values in the exponent of 1.3 because we thought they would decrease the score value without a steep decline. It could be another function with the same result. However, Figure 3 (previously it was Figure 2) demonstrates that, even for the worst parameters, the AUC values are clinically acceptable. The value “1.3” is defined by the “del1” parameter because it will always be less than 1.5 but more than 1.
We paraphrased the sentence (lines 350–351) as follows: “target region normalized the coverage values that were no more than 1.7”.
Only the k1, k2, k3, and k4 values are between 0 and 1. We have added the additional information for the k4 value (lines 357 and 367): “k4 has a value 1 if the exon deletion in question is covered by only one target region (except for the second exon which is frequently involved in the deletion of the BRCA1 second exon with promoter regions). Otherwise, k4 has a value of 0”.
- p. 11: “for the first and second CNV calling steps” you did not mention two different “steps” before. Do both or just one score have to be above the threshold?
We have corrected this phrase. It was about a two-step p-value evaluation, but not about calling CNVs/CNAs. Also, we added information about this procedure in the Results section (lines 155–160) and more details in the Materials and Methods section (lines 370–384).
“The p-value evaluation by the procedure described above is performed in two steps. In the first step, BRACNAC identifies CNVs/CNAs with a high score (by default, 9.9) and a low p-value (by default, 0.01). Then, it excludes such samples from the second step of the p-value evaluation to avoid their influence on the probability values for other potential CNVs/CNAs. We implemented this two-step p-value evaluation procedure to decrease the false-negative rate.”
However, we cannot start the description of the p-value evaluation process with the description of the two-step p-value evaluation in the Material and Methods section because we should initially describe how the p-value is calculated.
Thus, the main idea of this stage is to exclude obviously positive samples from the subsequent p-value evaluation. Such samples have many target region coverage values within the threshold (like in Supplementary Figure S2), and during the bootstrap analysis, these values would significantly increase the p-value for CNVs/CNAs of other samples.
We are grateful for this important question that helped us improve the description of the BRACNAC p-value calculation.
- At least one of the supplementary figures must be included in the manuscript itself to show an example of the output of the algorithm.
We have added Figure 2 with an example of the BRACNAC output plot.
Round 3
Reviewer 3 Report
Comments and Suggestions for Authors
The authors now make clear their methods.